# Associations of Maternal Urinary Concentrations of Phenols, Individually and as a Mixture, with Serum Biomarkers of Thyroid Function and Autoimmunity: Results from the EARTH Study

**DOI:** 10.3390/toxics11060521

**Published:** 2023-06-09

**Authors:** Glen McGee, Maximilien Génard-Walton, Paige L. Williams, T. I. M. Korevaar, Jorge E. Chavarro, John D. Meeker, Joseph M. Braun, Maarten A. Broeren, Jennifer B. Ford, Antonia M. Calafat, Irene Souter, Russ Hauser, Lidia Mínguez-Alarcón

**Affiliations:** 1Department of Statistics and Actuarial Science, University of Waterloo, Waterloo, ON N2L 3G1, Canada; glen.mcgee@uwaterloo.ca; 2Irset, EHESP, Inserm, Université de Rennes, 35700 Rennes, France; maximilien.genard-walton@univ-rennes.fr; 3Department of Biostatistics and Epidemiology, Harvard T.H. Chan School of Public Health, Boston, MA 02115, USA; paige@sdac.harvard.edu; 4Department of Internal Medicine and Academic Center for Thyroid Diseases, Erasmus University Medical Center, 3015 GD Rotterdam, GE, The Netherlands; t.korevaar@erasmusmc.nl; 5Departments of Epidemiology and Nutrition, Harvard T.H. Chan School of Public Health, Boston, MA 02115, USA; jchavarr@hsph.harvard.edu; 6Channing Division of Network Medicine, Harvard Medical School and Brigham and Women’s Hospital, Boston, MA 02114, USA; 7Department of Environmental Health Sciences, University of Michigan School of Public Health, Ann Arbor, MI 48109, USA; meekerj@umich.edu; 8Department of Epidemiology, Brown University, Providence, RI 02912, USA; joseph_braun_1@brown.edu; 9Laboratory of Clinical Chemistry and Haematology, Máxima Medical Centre, 5631 BM Veldhoven, De Run, The Netherlands; m.broeren@mmc.nl; 10Department of Environmental Health, Harvard T.H. Chan School of Public Health, Boston, MA 02115, USA; jford@hsph.harvard.edu; 11National Center for Environmental Health, Centers for Disease Control and Prevention, Atlanta, GA 30333, USA; aic7@cdc.gov; 12Vincent Obstetrics and Gynecology, Massachusetts General Hospital and Harvard Medical School, Boston, MA 02114, USA; isouter@mgh.harvard.edu (I.S.); rhauser@hsph.harvard.edu (R.H.); 13Departments of Epidemiology and Environmental Health, Harvard T.H. Chan School of Public Health, Boston, MA 02115, USA

**Keywords:** phenols, mixtures, BKMR, thyroid function

## Abstract

The associations between urinary phenol concentrations and markers of thyroid function and autoimmunity among potentially susceptible subgroups, such as subfertile women, have been understudied, especially when considering chemical mixtures. We evaluated cross-sectional associations of urinary phenol concentrations, individually and as a mixture, with serum markers of thyroid function and autoimmunity. We included 339 women attending a fertility center who provided one spot urine and one blood sample at enrollment (2009–2015). We quantified four phenols in urine using isotope dilution high-performance liquid chromatography–tandem mass spectrometry, and biomarkers of thyroid function (thyroid-stimulating hormone (TSH), free and total thyroxine (fT4, TT4), and triiodothyronine (fT3, TT3)), and autoimmunity (thyroid peroxidase (TPO) and thyroglobulin (Tg) antibodies (Ab)) in serum using electrochemoluminescence assays. We fit linear and additive models to investigate the association between urinary phenols—both individually and as a mixture—and serum thyroid function and autoimmunity, adjusted for confounders. As a sensitivity analysis, we also applied Bayesian Kernel Machine Regression (BKMR) to investigate non-linear and non-additive interactions. Urinary bisphenol A was associated with thyroid function, in particular, fT_3_ (mean difference for a 1 log unit increase in concentration: −0.088; 95% CI [−0.151, −0.025]) and TT_3_ (−0.066; 95% CI [−0.112, −0.020]). Urinary methylparaben and triclosan were also associated with several thyroid hormones. The overall mixture was negatively associated with serum fT_3_ concentrations (mean difference comparing all four mixture components at their 75th vs. 25th percentiles: −0.19, 95% CI [−0.35, −0.03]). We found no evidence of non-linearity or interactions. These results add to the current literature on phenol exposures and thyroid function in women, suggesting that some phenols may alter the thyroid system.

## 1. Introduction

Endocrine disrupting chemicals (EDCS) such as bisphenol A (BPA), benzophenone-3, parabens, and triclosan may interfere with the endocrine system, leading to detrimental health effects in both humans and wildlife [1]. BPA is found in synthetic polymers, building materials, thermal paper, toys, dental products, and food packaging [2]. Benzophenone-3 is widely used in cosmetic products as a sunscreen agent that absorbs and dissipates ultraviolet radiation [3]. Parabens, including methylparaben, butylparaben, and propylparaben, are used as food preservatives and shelf stabilizers [4,5] and in personal care products such as shampoos, creams [6], and pharmaceutical products [7,8]. Triclosan is an antimicrobial agent [9] that has been used in personal hygiene products such as mouthwashes, toothpastes, and hand sanitizers [10]. Despite these environmental phenols’ short elimination half-life (<24 h), exposures are repeated, episodic, and chronic [11,12,13]. Urine has been shown to be the optimal biological matrix for assessing exposure to these chemicals [14]. BPA, benzophenone-3, parabens, and triclosan have been detected in almost 100% of urine specimens from representative samples of the U.S. general population from the National Health and Nutrition Examination Survey (NHANES) [15], confirming that exposure to these chemicals is ubiquitous.

Animal studies have demonstrated an alteration of thyroid biomarkers after exposure to some environmental phenols. For example, exposures to methyl-, ethyl-, propyl-, isopropyl-, and isobutylparaben were associated with lower total thyroxine (TT_4_) concentrations in rat models [16]. In addition, benzophenone-3 was associated with higher levels of free T4 (fT_4_) and triiodothyronine (T_3_) and lower levels of thyroid-stimulating hormone (TSH) in female rats [17]. Additionally, exposure to triclosan decreased TT_4_ [18] and TT_3_ concentrations [18] in rats. Epidemiologic studies on BPA, benzophenone-3, parabens, and triclosan in relation to thyroid hormones in humans, however, have shown inconsistent findings [19,20,21,22,23,24]. Importantly, previous studies have not investigated these exposures as a mixture. Evaluating chemical mixtures is of public health importance because humans are exposed to multiple synthetic chemicals simultaneously, each of which may themselves be correlated or act synergistically [25,26]. To address this knowledge gap, we evaluated the urinary concentrations of BPA, benzophenone-3, parabens, and triclosan, individually and as a mixture, in relation to thyroid function and autoimmunity biomarkers among women attending a fertility center.

Human biomonitoring and environmental studies have demonstrated the presence of multiple chemical exposures, and thus it is of interest to evaluate the health effects of a group of chemicals that can biologically interact or share sources of exposure. As such, we analyzed phenol biomarkers as a chemical mixture. This may be problematic for a risk assessment, however, because some of the examined phenols may have different sources of exposure (e.g., parabens are mainly found in personal care products and bisphenol A, is, for example, found in food packaging, among others). Consequently, we also examined associations between individual phenol biomarkers and thyroid function, which is also important from a toxicological perspective.

## 2. Materials and Methods

### 2.1. Study Participants

Women in this study were enrolled in the Environment and Reproductive Health (EARTH) study, a prospective cohort designed to study environmental and dietary determinants of fertility among couples seeking fertility care at the Massachusetts General Hospital (MGH) Fertility Center [27]. Women aged 18–45 were eligible to participate. Among the N = 956 women enrolled in EARTH, this cross-sectional analysis included 339 women enrolled between 2009 and 2015 who provided a spot urine and a blood sample. We excluded 219 women without phenol data and 398 women lacking serum thyroid and autoimmunity biomarker data; as previously described, this included 133 women using thyroid-interfering medication (predominantly levothyroxine, methimazole, propylthiouracil, amiodarone, antipsychotics, anticonvulsants, or high-dose steroids) at study entry [28].

Each participant’s date of birth was collected at entry. Trained study staff measured weight and height, and body mass index (BMI) was computed as the ratio of weight to height squared (in kilograms per meter squared). After giving informed consent, participants completed questionnaires that were administered by study staff assessing their sociodemographic, lifestyle, and medical history at enrollment. Participants further completed a take-home questionnaire regarding their family, medical, reproductive, and occupational history, consumer products use, smoking history, and physical activity. Infertility was diagnosed according to the definitions of the Society of Assisted Reproductive Technology [29]. The study was approved by the Human Subject Committees of the Harvard T.H. Chan School of Public Health, MGH, and the Centers for Disease Control and Prevention (CDC).

### 2.2. Exposure Assessment

At enrollment, participants collected one spot urine sample in a sterile polypropylene specimen cup. Specific gravity (SG) was measured at room temperature using a handheld refractometer (National Instrument Company, Inc., Baltimore, MD, USA) calibrated with deionized water before each measurement. Rather than correcting for SG, we used unadjusted urinary phenol biomarker concentrations and adjusted for SG as a covariate in all statistical models to avoid bias [30,31]. The urine samples were stored at −80 °C and shipped frozen on dry ice overnight to the CDC for analysis. As previously described [32], we used online solid-phase extraction coupled with isotope dilution high-performance liquid chromatography–tandem mass spectrometry to measure urinary concentrations of six phenol biomarkers: BPA, benzophenone-3, triclosan, methylparaben, propylparaben, and butylparaben. Limits of detection (LOD) ranged from 0.1 to 1.0 µg/L, depending on the biomarker, and changed over the course of study. We excluded urinary butylparaben from the analysis because of the relatively low detection rate (57%), and we also excluded urinary propylparaben because it was highly correlated with methylparaben (r = 0.87) and had a lower detection rate. To accommodate concentrations below the LOD, we used a left-censored normal multiple imputation strategy (with m = 10 imputations), as described by Lapidus et al. [33], and implemented it via the mice doMIsaul “https://github.com/LilithF/doMIsaul (accessed on 1 April 2022) and multiLODmice (https://github.com/glenmcgee/multiLODmice (accessed on 1 April 2022)) packages in R.

Each analytical run included the following, in addition to study samples: a set of calibrators (prepared in methanol with commercially available analytical standards of the target analytes), reagent blanks (prepared in HPLC grade water), and high- and low-concentration quality control (QC) materials (made from urine spiked with known concentrations of the target analytes). Commercial sources of the analytical standards as well as details about the preparation of standards, blanks, and QCs have been reported before [34]. QC concentrations were evaluated using standard statistical rules [35]. If the QC samples failed statistical evaluation, all study samples in the run were re-extracted. The CDC’s analytical methods are public knowledge [34] and have been used to analyze tens of thousands of specimens since the early 2000s. These include samples collected as part of NHANES, which has provided the most comprehensive assessment of Americans’ exposure to phenols to date [36,37].

### 2.3. Outcome Assessment

From each participant, a single non-fasting blood sample was collected via venipuncture on the same day that the urine sample was collected. Serum samples were centrifuged, stored at −80 °C, and shipped on dry ice to the Department of Clinical Chemistry, Máxima Medical Center (Veldhoven, The Netherlands), to assess biomarkers of thyroid function and autoimmunity. The six outcomes of interest included serum concentrations of thyroid-stimulating hormone (TSH), free and total thyroxine (fT4, TT4), free and total triiodothyronine (fT3, TT3), thyroperoxidase antibody (TPOAb), and thyroglobulin antibodies (TgAb). Each concentration was quantified via electrochemoluminescence assays (Cobas^®^ e601 platform; Roche Diagnostics, Mannheim, Germany). Between-run coefficients of variation 2.1% for TSH, 3.5% for fT4, 3.8% for TT4, 3.8% for fT3, and 7.7% for TT3. Coefficients of variation were 12.4% for TPOAb at 33 IU/L and 7.1% at 100 IU/L, and 10.9% for TgAb at 76 IU/L and 8.6% at 218 IU/L. Clinical reference values were as follows: 0.4–4.0 mU/L for TSH, 10–24 pmol/L for fT4, 58–161 nmol/L for TT4, 3.5–6.5 pmol/L for FT3, and 0.9–2.8 nmol/L for TT3. TPOAb and TgAb concentrations were dichotomized to 1 (positive) if they were >35 IU/mL or >115 IU/L, respectively, and 0 otherwise (corresponding to manufacturer cutoffs).

### 2.4. Statistical Analysis

We summarized participants’ demographic and baseline reproductive characteristics via median and interquartile ranges (IQRs) or via counts and proportions (in %). We summarized distribution of urinary concentrations of phenol biomarkers via percentiles as well as geometric means and standard deviations (SDs). Due to right skewness, we log_e_-transformed urinary concentrations of phenol biomarkers and assessed pairwise correlations of urinary biomarker concentrations via Spearman correlation coefficients.

We assessed potential confounders using prior knowledge about biological relevance and descriptive analysis of the study sample. Variables were considered potential confounders if they were associated with urinary phenols biomarker concentrations and thyroid biomarkers but were not believed to lie on the causal pathway between exposure and outcome. All models—single-exposure and mixture models alike—were adjusted for age (years), BMI (kg/m^2^), and race (white vs. other), and were further adjusted for specific gravity (SG) to account for urine dilution.

We fit both single-exposure models (one model per exposure-outcome pair) as well as multi-exposure mixture models (one model per outcome that adjusted for all four phenols). We first fit linear models, regressing each continuous thyroid outcome on the natural log of phenol concentration(s), and we reported estimates and 95% confidence intervals (CIs) for mean difference in outcome for a 1 log unit (μg/L) increase in exposure biomarker concentration. We then fit additive models, in which the functional relationship between each phenol concentration and the thyroid outcome was allowed to be non-linear and was estimated non-parametrically via penalized splines. For additive models, we plotted estimated mean differences along with 95% CIs corrected for smoothness selection via restricted maximum likelihood. In the multi-exposure mixture models, we also estimated overall mixture associations, defined as mean differences in thyroid biomarkers for a simultaneous increase from 25th to 75th percentiles of all mixture exposures simultaneously. For the binary outcomes (TgAb and TPOAb), we fit generalized linear and additive models with a logit link, and we reported all associations on the odds ratio scale.

While the (generalized) additive models allowed for non-linear relationships, we further investigated non-additive interactions via Bayesian Kernel Machine Regression (BKMR) [26,38,39] in sensitivity analyses, and we plotted estimated exposure–response curves for component-wise associations and for pairwise interactions. Statistical analyses were performed using the mgcv [40] and mice [41] packages in R v4.0.2 (The R Foundation for Statistical Computing Platform). We reported 95% intervals and two-tailed 0.05 level tests where appropriate, but we emphasized consistency of findings across analyses rather than statistical significance.

## 3. Results

Among the 339 women in this study, the median (IQR) age was 34.0 (32.0, 38.0) years and the median BMI (IQR) was 23.2 (21.2, 26.2) kg/m^2^. Participants were predominantly white (83%), and 26% had ever smoked (Table 1). Median (IQR) serum concentrations of TSH, fT_4_, TT_4_, fT_3_, TT_3_, TgAb, and TPOAb were 1.85 (1.40, 2.60) mU/L, 15.5 (14.1, 16.7) pmol/L, 96.8 (86.5, 110) nmol/L, 4.80 (4.47, 5.21) pmol/L, 1.79 (1.58, 2.06) nmol/L, 15.9 (11.1, 23.3) IU/mL, and 12.0 (9.83, 16.0) IU/mL, respectively. TPOAb positivity and TgAb positivity were detected in 37 (11%) and 35 (10%) women. Serum thyroid function and autoimmunity biomarker concentrations were within normal ranges for healthy adult women. Detection frequencies for BPA, benzophenone-3, methylparaben, and triclosan were greater than 79% (Appendix A), and were similar to those reported in U.S. females from the general population [37]. The median urinary concentrations of these phenol biomarkers were also similar to those reported for females from the general U.S. population, except for benzophenone-3, which was higher in this study, as previously described [42]. The four urinary phenols were weakly correlated with one another (Spearman r = 0.17–0.32).

Results for the continuous thyroid biomarkers were similar between the linear single-exposure and multi-exposure mixture models (Figure 1; also see Appendix A), although CIs were wider in the mixture models. Urinary BPA was consistently associated with thyroid hormone biomarkers; there was a positive association with TSH and a negative association with fT_4_, TT_4_, fT_3_, and TT_3_ concentrations in the single-component analyses. Associations with fT_3_ (estimated mean difference per 1 log unit increase: −0.088; 95% CI [−0.151, −0.025]) and TT_3_ (−0.066; 95% CI [−0.112, −0.020]) remained significant in the mixture models. Urinary methylparaben was positively associated with TSH, fT_4_, and fT_3_ in the single-component analyses, but the evidence from mixture models was weaker. Urinary triclosan was weakly negatively associated with TSH, fT_3_, and TT_3_, but only the association with fT_3_ remained significant in mixture models (estimated mean difference per 1 log unit increase −0.041; [−0.067, −0.014]). Urinary benzophenone-3 was not related to any of the examined outcomes.

The additive models (Figure 2 and Appendix A) largely echoed the results of the linear models, with a few exceptions: BPA was only associated with fT_3_ and TT_3_; methylparaben was only associated with fT_3_ and fT_4_ in the mixture models; and triclosan was only associated with fT_3_. We observed no significant evidence of non-linear associations. The results of the BKMR models were similar to those of the additive models, albeit with more uncertainty (Appendix A); we found no evidence of interactions among mixture components (Appendix A).

Considering the entire mixture simultaneously (Table 2), we found some evidence of a negative overall mixture association with serum fT3 concentrations. Specifically, the estimated mean difference for an increase from the 25th to 75th percentiles of all exposures was −0.19 (95% CI (−0.35, −0.03)) in the linear mixture model, but there was more uncertainty in the additive mixture model (−0.26, 95% CI (−0.94, 0.41)).

In analyses for serum antibodies evaluated as binary outcomes, we observed a positive association between TgAb and methylparaben and a negative association between TgAb and triclosan in the single-component linear analyses (Figure 1). However, the results from the mixture models were not significant (Appendix A): ORs for 1 log unit increase in urinary concentration of 1.265 (95% CI [0.975, 1.641]) and 0.892 (95% CI [0.757, 1.051]), respectively, and no statistically significant associations were found in the generalized additive models (Figure 3 and Appendix A).

## 4. Discussion

We assessed the cross-sectional relationship between urinary concentrations of four phenol biomarkers and serum markers of thyroid function and thyroid autoimmunity among 339 subfertile women seeking fertility care in Boston (MA). In models assessing phenols individually, we observed that BPA was positively associated with TSH and negatively associated with fT_4_, TT_4_, fT_3_, and TT_3_ concentrations. Furthermore, methylparaben was positively associated with serum concentrations of TSH, fT_4_, fT_3_, and TgAb, and triclosan was negatively associated with TSH, fT_3_, TT_3_, and TgAb concentrations. Multi-exposure mixture models yielded similar estimates of individual associations, albeit with increased uncertainty, and we also observed a negative association for the mixture with serum fT_3_. These results contribute to the epidemiologic literature on gestational environmental exposures as determinants of thyroid function and autoimmunity.

In a subset of 317 women in this study cohort, we previously observed that urinary triclosan concentrations were negatively associated with specific serum thyroid function biomarkers [24]. Among 454 pregnant California women included in the Center for the Health Assessment of Mothers and Children of Salinas (CHAMACOS) study, researchers found a negative association between urinary triclosan and serum total T4, although this association did not remain after controlling for other chemicals. In a nested case–control sample of 439 pregnant women in Boston (MA), Aker et al. found lower plasma total T3 levels with higher urinary benzophenone-3, butylparaben, and triclosan [20]. The authors also reported a positive association between urinary methylparaben and plasma T3, as well as a negative association between urinary propylparaben and plasma-free T4. Among 602 pregnant women in the Puerto Rico Testsite for Exploring Contamination Threats (PROTECT) cohort, urinary bisphenol S was positively associated with serum fT_4_, whereas triclocarban was positively associated with serum TT3 and negatively associated with serum TSH [21]. Though no associations were found for other bisphenols and triclosan, urinary BPA was negatively associated with TT4 concentrations among 1996 women in the Swedish Environmental Longitudinal, Mother and child, Asthma and allergy study (SELMA), a population-based prospective pregnancy cohort [23]. However, triclosan was positively related to serum TSH in 3360 women from the second (2012–2014) Korean National Environmental Health Survey [22]. Among 181 pregnant women participating in the Health Outcomes and Measures of the Environment (HOME) study, researchers found no associations between urinary BPA and circulating thyroid hormones, including TSH, fT4, TT4, and TT3 [43]. Discrepancies in results between our study and the aforementioned studies evaluating phenol exposures individually in relation to thyroid function biomarkers may, in part, be due to the lack of consideration of other phenols when evaluating the associations. Other reasons may include different study populations and urinary phenol biomarker concentrations.

Among women in this study, exposure to selected phenols was associated with alterations in thyroid function, and some of them were suggestive of thyroid diseases. For example, urinary concentrations of BPA were associated with higher TSH and lower thyroid hormones, which is indicative of hypothyroidism [44]. Additionally, urinary methylparaben was positively associated with TSH, fT3, and fT4, which are observed when there is altered negative feedback and/or thyroid hormone resistance at the pituitary level. Finally, urinary triclosan was associated with lower TSH and lower T3. Increased circulating levels of both hormones are indicative of a non-thyroidal illness pattern or upregulation of type 3 deiodinase in peripheral tissues, including the pituitary.

This study includes women seeking fertility care, who are an important study population, as subfertile women have been demonstrated to be at a higher risk for thyroid disease [28,45,46]. A strength of this study is the use of several statistical methods to evaluate biomarker mixtures. While linear models are powerful, they make strong assumptions (i.e., no interactions and no non-linearity). By contrast, BKMR allows for non-linearity as well as high-order interactions among mixture components, but this flexibility decreases power. As a compromise, we also fit GAMs, which allow the estimation of flexible, non-linear relationships without committing to a fully non-parametric approach that allows for high-order interactions, which are difficult to estimate in small samples. Another strength is the multiple imputation strategy, which uses information about all observed data to impute non-detectable biomarker concentrations. Potential confounding bias was limited because all the participants attended a single medical center and were enrolled in an established cohort study with homogenous demographics. All serum and urine samples were collected and processed under a common protocol before determining thyroid function and thyroid autoimmunity biomarkers, and urinary phenol biomarker concentrations were quantified at the CDC using the analytical approach used in other studies including NHANES.

This study is not without limitations. First, the generalizability of the results to women in the general population is limited because this study is restricted to women attending a fertility center. Second, causality cannot be established given the cross-sectional nature of this study. Third, unmeasured confounding by factors affecting both phenol biomarkers and the thyroid system could cause spurious results. Fourth, non-differential exposure misclassification—because of the episodic exposure to the phenols examined and their relatively short biological half-lives, especially when only including one urine sample per woman—could bias estimates to the null [47,48]. However, we previously demonstrated in the EARTH study that a single urine sample can effectively reflect one’s exposure to BPA, triclosan and other short half-lived chemicals such as phthalates over several months [49,50].

## 5. Conclusions

In a sample of women attending a fertility center, we found that urinary phenols—specifically BPA, methylparaben, and triclosan—were associated with several serum markers of thyroid function and autoimmunity in both single- and multi-exposure mixture analyses. These findings contribute to the epidemiologic literature on environmental exposures during reproductive years as determinants of thyroid function and autoimmunity. As the thyroid system is strictly regulated and any effects on serum biomarker concentrations may have relevant physiological consequences, further studies should evaluate the long-term consequences as well as biological mechanisms (e.g., omics) to explain the observed findings.

## Figures and Tables

**Figure 1 toxics-11-00521-f001:**
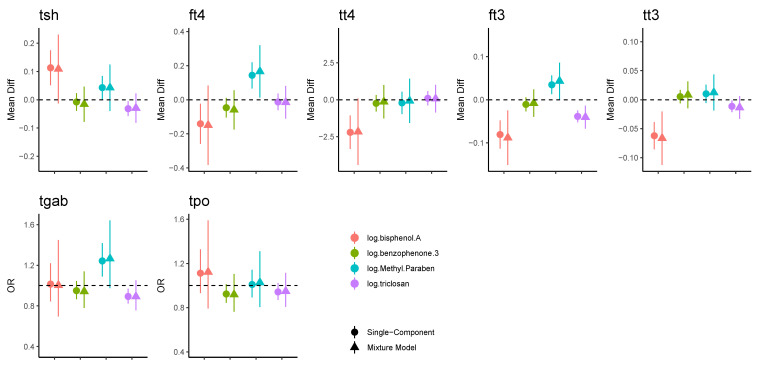
Associations between exposures and thyroid function in linear models. Estimates and corresponding 95% confidence intervals of mean differences (for continuous outcomes) and odds ratios (for binary outcomes) for a 1 log unit increase in concentration. Univariate corresponds to analyses with a single mixture component; multiple corresponds to mixture models with all four components. Models were adjusted for age (years), BMI (kg/m^2^), race (white vs. other), and specific gravity (SG).

**Figure 2 toxics-11-00521-f002:**
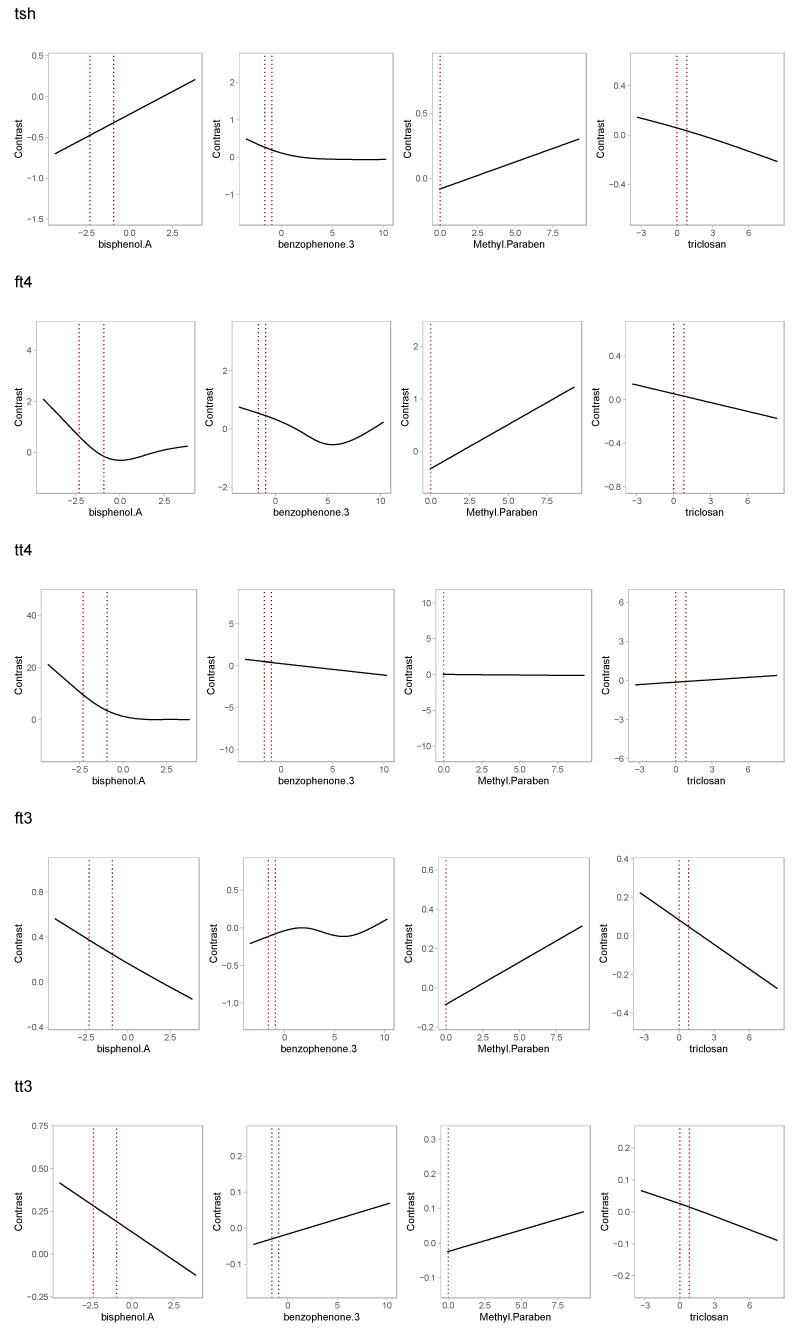
Mixture analysis: Additive model results for continuous outcomes. Curves represent estimated mean differences and corresponding 95% confidence intervals, compared to median log concentration. Each row corresponds to a different model. Models were adjusted for age (years), BMI (kg/m^2^), race (white vs. other), and specific gravity (SG).

**Figure 3 toxics-11-00521-f003:**
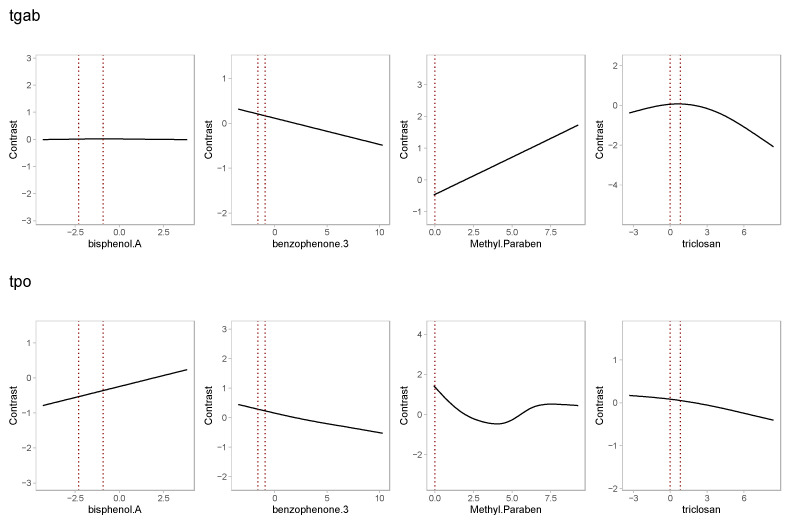
Mixture analysis: Generalized additive model results for binary outcomes. Curves represent estimated log odds ratios and corresponding 95% confidence intervals, compared to median log concentration. Each row corresponds to a different model. Models were adjusted for age (years), BMI (kg/m^2^), race (white vs. other), and specific gravity (SG).

**Table 1 toxics-11-00521-t001:** Demographics and reproductive characteristics as well as thyroid biomarkers (median (IQR) or N (%)) among 339 women in the Environment and Reproductive (EARTH) study.

Demographics	
Age, years	34.0 (32.0, 38.0)
White (race), N (%)	281 (83)
Body Mass Index, kg/m^2^	23.2 (21.2, 26.2)
Ever smoked, N (%)	90 (26)
Education, N (%)	
High school/some college	67 (20)
College graduate	99 (29)
Graduate degree	173 (51)
**Reproductive history**	
Initial infertility diagnosis, N (%)	
Male factor	80 (24)
Female factor	151 (44)
Unexplained	107 (32)
**Thyroid biomarkers**	
TSH (mU/L)	1.85 (1.40, 2.60)
Free T_4_ (pmol/L)	15.5 (14.1, 16.7)
Total T_4_ (nmol/L)	96.8 (86.5, 110)
Free T_3_ (pmol/L)	4.80 (4.47, 5.21)
Total T_3_ (nmol/L)	1.79 (1.58, 2.06)
TgAb positivity (>115 IU/mL), N (%)	37 (11)
TPOAb positivity (>35 IU/mL), N (%)	35 (10)

**Table 2 toxics-11-00521-t002:** Estimates of overall mixture association, comparing 75th to 25th percentiles of all mixture components simultaneously. Overall associations shown for multiple (generalized) linear models and (generalized) additive models. Est is estimated mean difference (or odds ratio for binary outcomes) comparing 75th to 25th percentiles of exposure biomarker concentration; 95% CI is corresponding confidence interval.

	Linear		Additive	
	Est	95% CI	Est	95% CI
TSH	0.11	(−0.20, 0.42)	0.13	(−0.70, 0.97)
fT_4_	−0.09	(−0.68, 0.49)	−0.13	(−1.36, 1.11)
TT_4_	−3.57	(−9.25, 2.12)	−3.14	(−6.72, 0.44)
fT_3_	−0.19	(−0.35, −0.03)	−0.26	(−0.94, 0.41)
TT_3_	−0.09	(−0.21, 0.02)	−0.09	(−0.58, 0.39)
TgAb	0.97	(0.37, 2.54)	0.90	(0.20, 4.07)
TPOAb	0.84	(0.33, 2.17)	1.29	(0.27, 6.25)

## Data Availability

The data are not publicly available due to privacy and confidentiality reasons.

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
