# Peer review of "Associations of Maternal Urinary Concentrations of Phenols, Individually and as a Mixture, with Serum Biomarkers of Thyroid Function and Autoimmunity: Results from the EARTH Study"

_toxics, 2023, doi:10.3390/toxics11060521_

Round 1

Reviewer 1 Report

Manuscript by McGee et al. aimed to explore the associations of maternal urinary concentrations of phenols, individually and as a mixture, with serum biomarkers of thyroid function and autoimmunity by using the results from the EARTH Study. The methodology used in this research is appropriate for the hypotheses tested and the conclusions discussed are consistent with experimental data. I have only a few minor comments which should be addressed to enhance the overall quality of the paper.

In the abstract, separation method should be included alongside the determination method (isotope dilution-high-performance liquid chromatography-tandem mass spectrometry instead of only mass spectrometry).

In the M&M section, the authors have stated that, along with the study samples, each analytical run included a set of calibrators, reagent blanks, and high- and low-concentration quality control (QC) materials. All of these should be listed and explained in more detail. What did reagent blanks consist of? Manufacturer of quality control materials and calibrators?

Author Response

Manuscript by McGee et al. aimed to explore the associations of maternal urinary concentrations of phenols, individually and as a mixture, with serum biomarkers of thyroid function and autoimmunity by using the results from the EARTH Study. The methodology used in this research is appropriate for the hypotheses tested and the conclusions discussed are consistent with experimental data. I have only a few minor comments which should be addressed to enhance the overall quality of the paper.

Response: Thank you for the positive feedback and constructive comments.

In the abstract, separation method should be included alongside the determination method (isotope dilution-high-performance liquid chromatography-tandem mass spectrometry instead of only mass spectrometry).

Response: In the updated abstract, mass spectrometry has been now replaced for isotope dilution-high-performance liquid chromatography-tandem mass spectrometry as suggested.

“We quantified four phenols in urine using isotope dilution-high-performance liquid chromatography-tandem mass spectrometry, and biomarkers of thyroid function [thyroid stimulating hormone (TSH), free and total thyroxine (fT4, TT4), and triiodothyronine (fT3, TT3)], and autoimmunity [thyroid peroxidase (TPO) and thyroglobulin (Tg) antibodies (Ab)] in serum using electrochemoluminescence assays.”

In the M&M section, the authors have stated that, along with the study samples, each analytical run included a set of calibrators, reagent blanks, and high- and low-concentration quality control (QC) materials. All of these should be listed and explained in more detail. What did reagent blanks consist of? Manufacturer of quality control materials and calibrators?

Response: Per Reviewer suggestion, we have now included more detail in methods section regarding urinary phenol quantification.

“Each analytical run included, in addition to study samples: a set of calibrators (prepared in methanol with commercially available analytical standards of the target analytes), reagent blanks (prepared in HPLC grade water), and high- and low-concentration quality control (QC) materials (made from urine spiked with known concentrations of the target analytes). Commercial sources of the analytical standards as well as details about the preparation of standards, blanks and QCs have been reported before [34].

Reviewer 2 Report

The manuscript entitled "Title: Associations of maternal urinary concentrations of phe-2 nols, individually and as a mixture, with serum biomarkers of 3 thyroid function and autoimmunity: results from the EARTH 4 Study", is really a very interesting and actual topic. Endocrine disrupting chemicals are emerging contaminants and consumers are exposed continuosly to cocktail of compounds. The manuscript is really interesting but it is too important a current issue to have such a short introduction. In my opinion, the authors should go deeper into the introduction of the problem of exposure to mixtures of compounds, and its difficulty in risk assessment. By the other hand, reported data are really interesting and the discussion is well done, but the conclusion is weak, the authors should deepen it based on the results obtained and include futur lines of research.

Author Response

The manuscript entitled "Title: Associations of maternal urinary concentrations of phenols, individually and as a mixture, with serum biomarkers of 3 thyroid function and autoimmunity: results from the EARTH 4 Study", is really a very interesting and actual topic. Endocrine disrupting chemicals are emerging contaminants and consumers are exposed continuously to cocktail of compounds. The manuscript is really interesting but it is too important a current issue to have such a short introduction. In my opinion, the authors should go deeper into the introduction of the problem of exposure to mixtures of compounds, and its difficulty in risk assessment. By the other hand, reported data are really interesting and the discussion is well done, but the conclusion is weak, the authors should deepen it based on the results obtained and include future lines of research.

Response: We agree with the Reviewer and these two points have been now clarified in the introduction and conclusions, respectively, as suggested.

Human biomonitoring and environmental studies have demonstrated the presence of multiple chemical exposures, and thus it is of interest to evaluate the health effects of a group of chemicals that can biologically interact or share sources of exposure. As such, we analyzed phenol biomarkers as a chemical mixture. This may be problematic for risk assessment, however, because some of the examined phenols may have different sources of exposure (e.g. parabens are mainly found in personal care products and bisphenol A are, for example, found in food packaging among others). Consequently, we also examined associations between individual phenol biomarkers and thyroid function, which is also important from a toxicological perspective.

.”

“As the thyroid system is strictly regulated and any effects on serum biomarker concentrations may have relevant physiological consequences, further studies should evaluate long-term consequences as well as biological mechanisms (e.g. omics) explaining the observed findings.”